# Stroke Prediction with Machine Learning Methods among Older Chinese

**DOI:** 10.3390/ijerph17061828

**Published:** 2020-03-12

**Authors:** Yafei Wu, Ya Fang

**Affiliations:** 1The State Key Laboratory of Molecular Vaccine and Molecular Diagnostics, School of Public Health, Xiamen University, Xiamen 361102, China; wyfyyahcx@163.com; 2Key Laboratory of Health Technology Assessment of Fujian Province, School of Public Health, Xiamen University, Xiamen 361102, China; 3National Institute for Data Science in Health and Medicine, Xiamen University, Xiamen 361102, China

**Keywords:** stroke, imbalanced data, machine learning, prediction

## Abstract

Timely stroke diagnosis and intervention are necessary considering its high prevalence. Previous studies have mainly focused on stroke prediction with balanced data. Thus, this study aimed to develop machine learning models for predicting stroke with imbalanced data in an elderly population in China. Data were obtained from a prospective cohort that included 1131 participants (56 stroke patients and 1075 non-stroke participants) in 2012 and 2014, respectively. Data balancing techniques including random over-sampling (ROS), random under-sampling (RUS), and synthetic minority over-sampling technique (SMOTE) were used to process the imbalanced data in this study. Machine learning methods such as regularized logistic regression (RLR), support vector machine (SVM), and random forest (RF) were used to predict stroke with demographic, lifestyle, and clinical variables. Accuracy, sensitivity, specificity, and areas under the receiver operating characteristic curves (AUCs) were used for performance comparison. The top five variables for stroke prediction were selected for each machine learning method based on the SMOTE-balanced data set. The total prevalence of stroke was high in 2014 (4.95%), with men experiencing much higher prevalence than women (6.76% vs. 3.25%). The three machine learning methods performed poorly in the imbalanced data set with extremely low sensitivity (approximately 0.00) and AUC (approximately 0.50). After using data balancing techniques, the sensitivity and AUC considerably improved with moderate accuracy and specificity, and the maximum values for sensitivity and AUC reached 0.78 (95% CI, 0.73–0.83) for RF and 0.72 (95% CI, 0.71–0.73) for RLR. Using AUCs for RLR, SVM, and RF in the imbalanced data set as references, a significant improvement was observed in the AUCs of all three machine learning methods (*p* < 0.05) in the balanced data sets. Considering RLR in each data set as a reference, only RF in the imbalanced data set and SVM in the ROS-balanced data set were superior to RLR in terms of AUC. Sex, hypertension, and uric acid were common predictors in all three machine learning methods. Blood glucose level was included in both RLR and RF. Drinking, age and high-sensitivity C-reactive protein level, and low-density lipoprotein cholesterol level were also included in RLR, SVM, and RF, respectively. Our study suggests that machine learning methods with data balancing techniques are effective tools for stroke prediction with imbalanced data.

## 1. Introduction

Stroke, accounting for 10% [1] of total deaths and 5% [2] of all disability-adjusted life-years worldwide, has posed a serious threat to population health, especially in developing countries with a low or moderate income [3]. With the acceleration of population aging, China has faced the biggest burden of stroke in recent years [4,5]. Stroke is the leading cause of death in China [6]. Predictive studies have found that the estimated lifetime stroke risk for individuals aged 25 years or more in China is expected to reach very high levels, and this will be a major concern for the prevention and management of stroke in the future [7].

To prevent an epidemic of stroke, accurate and flexible assessment tools are extremely important, especially in China where the incidence of stroke is high. There are several stroke prediction tools in China. However, traditional methods, such as the Cox proportional hazard model [6,8], are unable to effectively explore the complex non-linear relationships in data. Machine learning methods can learn complex structures by incorporating numerous variables with high dimensional data [9]. Excellent performance of these methods has been validated in health service [10] and health outcomes studies [11]. Regularized logistic regression (RLR), as the fundamental and most commonly used machine learning method, is a generalized linear regression model for probability analysis. It is usually used as the reference for performance comparisons between different machine learning methods in prediction [12]. Support vector machine (SVM) is a maximum interval classification method first proposed by Vapnik in 1995 [13]. It can obtain global optimal solutions with the assurance of its perfect theoretical basis and structural risk minimization criteria. In addition, SVM can efficiently handle complicated non-linear classification issues through kernel functions, so it is also a widely used classical disease prediction method [14]. Random forest is a commonly used ensemble learning method proposed by Leo Breiman et al. in 2001 [15]. It is designed with a combination of multiple decision trees, and it enables random selection for variables in model construction, which makes it more powerful for dealing with over-fitting in prediction [16].

Notably, the issue of imbalanced data, i.e., when the sample sizes between different classes are imbalanced, is unavoidable in prediction research [17]. Traditional data processing and prediction methods are more inclined toward representing the majority class and ignore the characteristics of the minority class, leading to a bias in results. Random over-sampling (ROS), random under-sampling (RUS), and synthetic minority over-sampling technique (SMOTE) are the three most commonly used balancing techniques for imbalanced data [17]. Specifically, ROS is the simplest method of over-sampling. The essence of ROS is to randomly copy samples in the minority class, therefore achieving consistency with the samples in the majority class. RUS is a commonly used method of under-sampling. It randomly removes samples from the majority class until the samples are equivalent to each other in the minority and majority classes. For synthetic sampling, SMOTE is a powerful tool for dealing with imbalanced data [18]. It creates new artificial samples based on similarities between minority samples. Several studies have shown significant improvements in performance after data balancing [19,20,21,22,23]. For instance, Zhang et al. used a support vector machine for the classification of breast cancer with random over-sampling combined with k-means for the selection of informative samples; the results showed much greater areas under the receiver operating characteristic curves (AUCs) with balanced data [20]. Similarly, researchers have also used the SMOTE technique to process imbalanced data in Parkinson’s disease, diabetes, and vertebral column pathologies, and results have shown that the accuracy, F-measure, and AUC all improved significantly for the machine learning methods used in study with the SMOTE balancing technique [22]. Furthermore, several machine learning methods with a wide range of ROS, RUS, and combination balancing techniques were compared with each other for dealing with imbalanced data on prostate cancer, and significant improvements in accuracy were observed in the AdaBoost methods with the SVMSMOTE balancing technique [23]. Therefore, imbalanced data require further consideration before prediction. To the best of our knowledge, previous studies have not discussed this issue in depth for stroke prediction.

Here, our study reports a stroke prediction model based on the three most commonly used machine learning methods (RLR, SVM, and RF), using Chinese Longitudinal Healthy Longevity Study (CLHLS) data. Simultaneously, the performance of machine learning methods with different data balancing techniques was further investigated.

## 2. Methods

### 2.1. Data Source

The data in this study were obtained from the Chinese Longitudinal Healthy Longevity Study (https://opendata.pku.edu.cn/dataverse/CHADS), which included elderly individuals from 23 provinces in China. This longitudinal study mainly covered demographic, lifestyle, and clinical information. The data quality is among the highest for similar surveys worldwide and has been widely recognized and used in academia. We used the latest available data in CLHLS for stroke prediction, with predictors and outcome data originating from 2012 and 2014, respectively. Specifically, 2439 participants were included in the baseline (2012). Participants with stroke who were missing values for the stroke variable and the predictive variable, and with age less than 60 years old, were excluded from the baseline. Participants with missing values for the outcome variable in 2014 were also excluded.

### 2.2. Predictors and Data Preprocessing

In all, 15 variables in three categories in 2012 were selected as predictors in this study, including demographic variables, such as sex, age, and comorbidities (hypertension, diabetes, and heart disease); lifestyle variables, such as smoking and drinking; and clinical variables including systolic blood pressure (SBP), diastolic blood pressure (DBP), high-sensitivity C-reactive protein (hsCRP), blood glucose (GLU), high-density lipoprotein cholesterol (HDLC), low-density lipoprotein cholesterol (LDLC), triglyceride (TG), and uric acid (UA). All the predictors are listed in Appendix A.

For data preprocessing, continuous variables were converted into categorical variables in analysis, and one-hot encoding was used for predictors. Considering the imbalanced classes between the stroke and non-stroke (approximate 1:19) participants, we used ROS, RUS, and SMOTE techniques to process the original data and achieve balance between the two classes. The details of data balancing methods are described in the Appendix A. We did not conduct imputation for missing data because the missing rate of predictors was not large (≤5.00%). 

### 2.3. Outcome Definition

The outcome was defined as stroke with two classes (stroke and non-stroke) in this study, and it was obtained from the following two-choice question: “*Are you suffering from stroke (yes/no)?*” It was self-reported by the elderly patients or their family members. The outcome data were obtained from CLHLS in 2014.

### 2.4. Model Derivation and Validation

In this study, 10-fold cross-validation was applied for model derivation and validation. In brief, 10-fold cross-validation means that all the data are divided into 10 equal parts. Then, 9 of those parts are used for training, and the remaining part is used as a testing set. Finally, the average value of the results is measured to obtain a more stable performance [24]. The details of cross-validation are explained in Appendix A. We used accuracy, sensitivity, specificity, and AUC to evaluate the performance of machine learning methods in our study. 

### 2.5. Statistical Analysis

Continuous variables are presented as mean ± SD (normal distribution) and as medians with inter-quartile range (IQR, skewed distribution). Categorical variables are presented as percentages. *T*-test, Wilcoxon rank sum test, and $\chi^{2}$ test were used for statistical comparison between stroke and non-stroke populations. In order to obtain the *p*-values of the AUC comparisons between different methods, we looped through the code 1000 times, and then a *Z*-test was used to calculate the *p*-value according to the following formula. Here, $\bar{x1}$ and $\bar{x2}$ represent the mean values of AUC for different machine learning methods. s1, s2 and n1, n2 indicate the standard deviations and sample sizes, respectively. *P*-value of ≤ 0.01 is defined as |Z|≥2.58, *p*-value of ≤0.05 is defined as |Z|≥1.96, and *p*-value of >0.05 is defined as |Z|<1.96.
$$Z=\frac{\bar{x1}-\bar{x2}}{\sqrt{\frac{s1}{n1}+\frac{s2}{n2}}}$$

The above analyses were performed using SPSS (version 22 for Windows). The derivation and validation of machine learning methods were completed by Python 3.7 with the Scikit-learn toolkit. A two-sided *p*-value of <0.05 was considered statistically significant.

## 3. Results

### 3.1. Baseline Characteristics

In this study, 2439 participants were included in 2012. After excluding 1109 participants with missing data (293 participants missing for predictors, 36 and 780 participants without stroke status in 2012 and 2014, respectively), 183 participants with stroke in 2012, and 16 participants aged less than 60 years, finally, 1131 participants were included in study. Among the participants, 56 were stroke patients and 1075 were non-stroke participants, as shown in Figure 1. The stroke prevalence was 4.95% for the elderly population in 2014, with men experiencing a much higher prevalence than women (6.76% vs. 3.25%).

*T*-test, Wilcoxon rank sum test, and $\chi^{2}$ test were used for statistical comparisons of predictors between stroke and non-stroke populations in 2012 (Table 1). The mean ages of stroke and non-stroke populations were 79.00 and 83.00 years, respectively. The proportion of males in the stroke population was higher than that in the non-stroke population (*p* < 0.05). With regard to clinical variables, a significant difference was observed in GLU and HDLC between the two groups (*p* < 0.05). With regard to comorbidities, the prevalence of hypertension was higher in the non-stroke population than in the stroke population (*p* < 0.05). No significant differences were observed in other variables between the two groups. 

### 3.2. Performance of Machine Learning Methods for Analyzing Imbalanced Data

The ratio of stroke to non-stroke population in this study was approximately 1:19, indicating a severe imbalance between the two classes. Therefore, data balancing operations were conducted for imbalanced data. Subsequently, RLR, SVM, and RF were used to construct prediction models for the imbalanced data set as well as the three new data sets (ROS-balanced, RUS-balanced, and SMOTE-balanced data sets). Accuracy, sensitivity, specificity, and AUC were measured to evaluate model performance (Table 2). When machine learning methods were used in the imbalanced data set, we observed high accuracy but extremely low sensitivity (almost 0.00) and AUCs (almost 0.50) in the validation data set. In contrast, the overall performance of machine learning methods improved significantly after the data balancing process. Although the accuracy and specificity declined slightly, the sensitivity and AUC improved greatly, with sensitivity increasing from approximately 0.00 to 0.78, and AUC increasing from approximately 0.50 to 0.72.

### 3.3. Performance Comparison between Different Data Balancing Methods

To explore the influence of data balancing methods on the performance of machine learning methods, we compared the AUCs in the ROS-balanced, RUS-balanced, and SMOTE-balanced data sets with those in the imbalanced data set for all the three machine learning methods. The results are shown in Figure 2. The performance of RLR significantly improved after the data balancing process compared with that before balancing (*p* < 0.01), and similar results were also observed for SVM and RF (*p* < 0.01).

### 3.4. Performance Comparison between Different Machine Learning Models

Comparisons of AUCs were also conducted between different machine learning methods. Considering the AUC of logistic regression as a reference, we compared the AUCs of SVM and RF with those of RLR in the imbalanced data set as well as the three balancing data sets. The results show that for the imbalanced data set, the performance of RF was better than that of RLR (*p* < 0.01), but the AUC of RF was relatively low (0.52 (95% CI, 0.51–0.53)), and there was no significant difference between SVM and RLR (*p* > 0.05) in the imbalanced data set (Figure 3). For the ROS-balanced data set, the results for SVM were much better than those for RLR (SVM: 0.71 (95% CI, 0.68–0.74); logistic: 0.67 (95% CI, 0.65–0.69), *p* < 0.01), and there was no significant difference between RF and RLR (*p* > 0.05). In the RUS-balanced and SMOTE-balanced data sets, the performance of both SVM and RF were equivalent to that of RLR (*p* > 0.05).

In general, the synthesized performance of machine learning methods improved greatly after the data balancing process, especially in terms of sensitivity and AUC. As shown in Figure 4, after SMOTE balancing, the AUC for RLR reached the maximum value 0.72 (95% CI, 0.71–0.73), and those for SVM and RF both reached 0.71 (95% CI, 0.70–0.72). The performance of other balancing methods (ROS and RUS) also improved considerably (Appendix A).

### 3.5. Selection of Important Predictors for the Stroke Prediction Model

Based on the SMOTE-balanced data set, we selected the top five important variables in three machine learning methods for stroke prediction after calculating the importance of each variable. Specifically, for RLR, the standardized regression coefficient was used to select the top five important variables. Similarly, the coefficient of each variable in the hyperplane equation was calculated for SVM, then the top five variables were selected based on the coefficients. For RF, the average contribution (Gini coefficient) was calculated for each variable based on all the trees in random forest, then the top five variables were selected with Gini coefficients in ascending order. The results are shown in Table 3; among them, sex, hypertension, and UA levels were common predictors for all three machine learning methods, and GLU level was an important predictor for both RLR and RF. In addition, drinking was included in RLR, age and hsCRP level were included in SVM, and LDLC level was included in RF.

## 4. Discussion

The prevalence of stroke has been extremely high in the elderly in China. With frequent exposure to risk factors such as hypertension [25], the prevention and management of stroke will be a major challenge in China in the near future.

Data in this study were quite imbalanced between the two classes (stroke vs. non-stroke), approximately 1:19. The performance of machine learning methods was poor with the imbalanced data set, as expected. Meanwhile, the synthesized performance of these prediction methods improved significantly after the data balancing process, indicating that it is unwise to perform prediction with imbalanced data. With data balancing techniques, severely imbalanced classes could be effectively avoided, which is crucial for accurate prediction [17]. 

The AUC for RLR, SVM, and RF in the ROS-balanced, RUS-balanced, and SMOTE-balanced data sets improved greatly compared with that in the imbalanced data set, respectively. This emphasizes the importance of data balancing techniques. Only the AUC for RF in the imbalanced data set and the AUC for SVM in the ROS-balanced data set showed improvement compared with RLR, and no significant differences were observed between the remaining models and RLR. This demonstrated that the performance of machine learning methods largely depended on the application scenario [26]. RLR, as a classical machine learning method with its flexibility and simplicity [27], showed quite a good performance in our study. SVM is also a powerful machine learning method [13], and it also showed excellent performance in the ROS-balanced data set in our study. RF is a typical representative for ensemble learning, which majorly aggregates the results of multiple classifiers to achieve more accurate predictions [15]. Although the original data set was quite imbalanced, RF still performed better than RLR.

The important predictors found in our study were consistent with those reported in previous studies. Sex, hypertension, and UA were the common factors included in all the three machine learning methods. Sex was the most important factor for stroke prediction in our study and was categorized into demographic variables, as validated in a previous study [28]. Hypertension was also an important predictor [29,30], and the prevention of hypertension is also an urgent task given its high prevalence and adverse influence on stroke in China. UA was found to be an independent predictor of early death in stroke patients [31]. GLU was an important factor common to RLR and RF. A previous study showed that impairment in GLU regulation was an important sign in the prognosis of stroke [32]. Drinking, hsCRP, and LDLC, as included in several previous studies [33,34,35], were also relatively important factors for RLR, SVM, and RF in our study, respectively.

The advantages of this study are as follows. First, the data used in this study were obtained from prospective cohorts with high-quality data representing the elderly in China, which could provide guidance for stroke prevention and management in the population aged 60 years and older. Second, the predictors used in our study were comprehensive, including demographic, lifestyle, and clinical variables, which allowed us to explore the relationship between risk factors and stroke from multiple perspectives. Finally, we conducted in-depth discussions on imbalanced data and compared the influence of three common data balancing techniques on the performance of machine learning methods, providing references for future studies. 

Our study also had several limitations. The outcome variable of this study was self-reported stroke; therefore, there may be some subjective bias. Furthermore, limited by data availability, the population included in our study was not large enough. Meanwhile, almost half of the participants were excluded due to the large proportion of missing outcome and predictive variables; this may also bring some uncertainty to our results. Furthermore, with the development of data balancing techniques, more methods are emerging; however, we only discussed the three most commonly used methods (ROS, RUS, and SMOTE) in this study. Finally, we only performed internal validation for our methods, and external validation in large populations is needed in future studies.

## 5. Conclusions

With the help of data balancing techniques, machine learning methods have the potential to improve the predictive performance with imbalanced data. We suggest that imbalanced data require proper treatment before prediction in future studies. 

## Figures and Tables

**Figure 1 ijerph-17-01828-f001:**
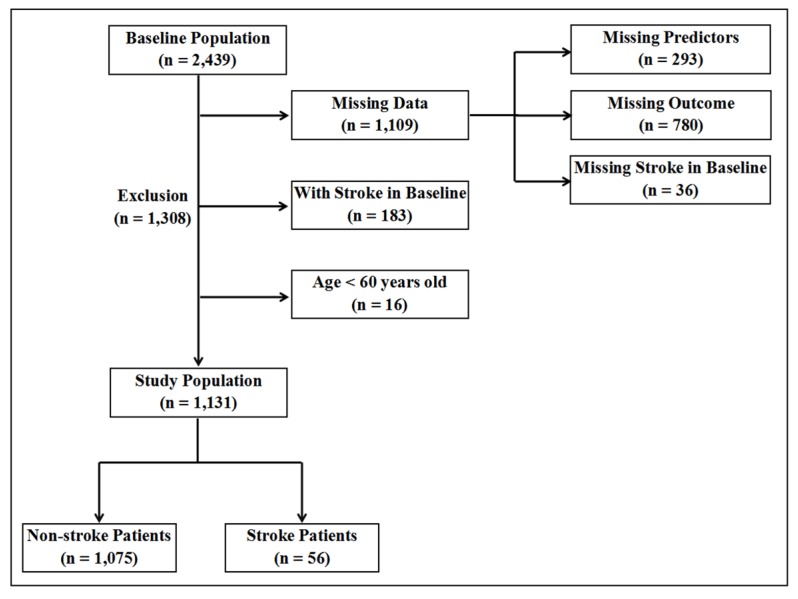
A flow chart for study population selection.

**Figure 2 ijerph-17-01828-f002:**
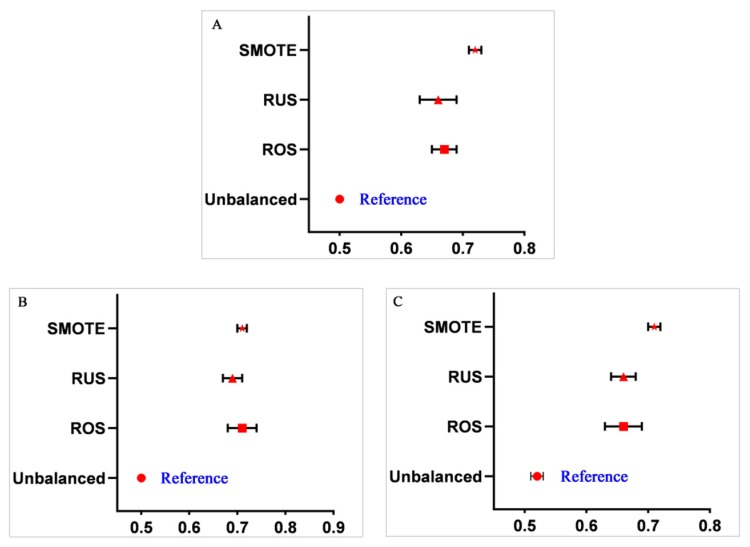
Comparison of AUCs for machine learning methods in different data sets. (**A**) Regularized logistic regression (RLR); (**B**) support vector machine (SVM); and (**C**) random forest (RF).

**Figure 3 ijerph-17-01828-f003:**
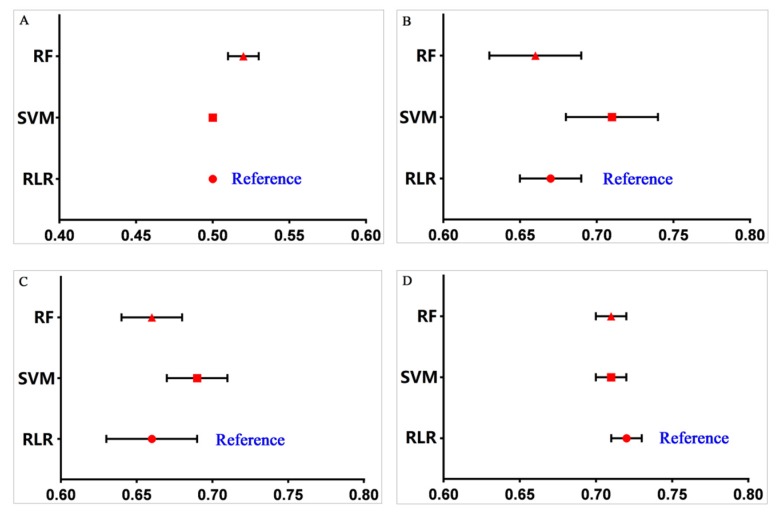
Comparison of AUCs between machine learning methods within each data set. (**A**) Imbalanced data; (**B**) random over-sampling (ROS); (**C**) random under-sampling (RUS); and (**D**) synthetic minority over-sampling technique (SMOTE).

**Figure 4 ijerph-17-01828-f004:**
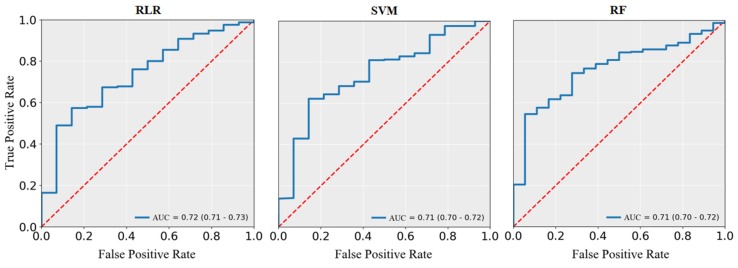
The receiver operating characteristic curves for machine learning methods in SMOTE-balanced data set.

**Table 1 ijerph-17-01828-t001:** Characteristics of the study population in 2012.

	Stroke (n = 56)	Non-Stroke (n = 1075)	*p*-value
Median age (IQR), y	79.00 (71.25~86.00)	83.00 (72.00~93.00)	0.08
Men, n (%)	37 (6.76%)	510 (93.24%)	0.01
Women, n (%)	19 (3.25%)	565 (96.75%)	
Median hsCRP (IQR), mg/L	0.85 (0.52~3.27)	0.77 (0.35~1.95)	0.14
Median GLU (IQR), mmol/L	4.65 (4.15~5.14)	4.46 (3.71~5.12)	0.03
Median HDLC (IQR), mmol/L	1.10 (0.93~1.41)	1.25 (1.05~1.51)	0.01
mean ± SD LDLC, mmol/L	2.56 ± 0.84	2.61 ± 0.80	0.66
Median TG (IQR), mmol/L	0.83 (0.59~1.13)	0.82 (0.60~1.17)	0.80
mean ± SD UA, umol/L	268.98 ± 86.31	287.01 ± 87.52	0.13
Smoker, n (%)	9 (3.83%)	226 (96.17%)	0.37
Non-smoker	47 (5.25%)	849 (94.75%)	
Drinker, n (%)	6 (2.83%)	206 (97.17%)	0.11
Non-drinker, n (%)	50 (5.44%)	869 (94.56%)	
Hypertension, n (%)	21 (7.29%)	267 (92.71%)	0.03
Non-hypertension, n (%)	35 (4.15%)	808 (95.85%)	
Diabetes, n (%)	2 (7.69%)	24 (92.31%)	0.85
Non-diabetes, n (%)	54 (4.89%)	1051 (95.11%)	
Heart disease, n (%)	6 (7.69%)	72 (92.31%)	0.38
Non-heart disease, n (%)	50 (4.75%)	1003 (95.25%)	
mean ± SD SBP, mmHg	142 ± 20	141 ± 24	0.88
Median DBP (IQR), mmHg	80 (74~90)	80 (71~89)	0.50

hsCRP indicates high-sensitivity C-reactive protein; GLU, blood glucose; HDLC, high-density lipoprotein cholesterol; LDLC, low-density lipoprotein cholesterol; TG, triglyceride; UA, uric acid; SBP, systolic blood pressure; and DBP, diastolic blood pressure.

**Table 2 ijerph-17-01828-t002:** Performance of machine learning methods in different data sets.

Model	Balancing Methods	Accuracy (95% CI)	Sensitivity (95% CI)	Specificity (95% CI)	AUC (95% CI)
RLR	–	0.95 (0.95–0.95)	0.00 (0.00–0.00)	1.00 (1.00–1.00)	0.50 (0.50–0.50)
SVM	–	0.95 (0.94–0.96)	0.00 (0.00–0.00)	1.00 (1.00–1.00)	0.50 (0.50–0.50)
RF	–	0.90 (0.89–0.91)	0.09 (0.06–0.12)	0.94 (0.93–0.95)	0.52 (0.51–0.53)
RLR	ROS	0.61 (0.59–0.63)	0.75 (0.71–0.79)	0.60 (0.58–0.62)	0.67 (0.65–0.69)
SVM	ROS	0.67 (0.65–0.69)	0.75 (0.70–0.80)	0.67 (0.65–0.69)	0.71 (0.68–0.74)
RF	ROS	0.68 (0.66–0.70)	0.63 (0.57–0.69)	0.68 (0.66–0.70)	0.66 (0.63–0.69)
RLR	RUS	0.57 (0.55–0.59)	0.76 (0.69–0.83)	0.56 (0.54–0.58)	0.66 (0.63–0.69)
SVM	RUS	0.62 (0.59–0.65)	0.77 (0.71–0.83)	0.61 (0.57–0.65)	0.69 (0.67–0.71)
RF	RUS	0.56 (0.53–0.59)	0.78 (0.73–0.83)	0.55 (0.52–0.58)	0.66 (0.64–0.68)
RLR	SMOTE	0.70 (0.68–0.72)	0.75 (0.72–0.78)	0.69 (0.67–0.71)	0.72 (0.71–0.73)
SVM	SMOTE	0.72 (0.68–0.76)	0.70 (0.66–0.74)	0.72 (0.68–0.76)	0.71 (0.70–0.72)
RF	SMOTE	0.78 (0.77–0.79)	0.62 (0.58–0.66)	0.79 (0.77–0.81)	0.71 (0.70–0.72)

RLR indicates regularized logistic regression; SVM, support vector machine; RF, random forest; ROS, random over-sampling; RUS, random under-sampling; SMOTE, synthetic minority over-sampling technique; and AUC, area under the receiver operating characteristic curve.

**Table 3 ijerph-17-01828-t003:** The top five important factors in the three machine learning methods.

Rank	Machine Learning Methods		
	RLR	SVM	RF
1	Sex	UA	Sex
2	Drinking	Sex	LDLC
3	Hypertension	Hypertension	GLU
4	GLU	Age	Hypertension
5	UA	hsCRP	UA

GLU indicates blood glucose; UA, uric acid; hsCRP, high-sensitivity C-reactive protein; LDLC, low-density lipoprotein cholesterol; RLR, regularized logistic regression; SVM, support vector machine; and RF, random forest.

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
