# Peer review of "Stroke Prediction with Machine Learning Methods among Older Chinese"

_ijerph, 2020, doi:10.3390/ijerph17061828_

Round 1
Reviewer 1 Report
Authors apply 3 methods to address the imbalance data and apply 3 machine learning methods to predict stroke. I suggest the following improvements.
Major:
- The title is overselling and can be made clear that this study is investigating ML (machine learning) methods to predict stroke.
- Introduction and literature review is very small, since one part of the study is to address the important issue of imbalanced datasets in health system so authors should review wider studies e.g. doi: 10.3390/data4030129 apply and compared 17 imbalance methods and 7 ML to predicting prostate cancer.
- Why are you only testing 3 ML methods, discuss.
Minors:
- In Fig 2, as all results are p<0.01 the ** on the graphs to indicate p<0.01 were a bit confusing.
- Info in Fig 4 may be shown in a single graph with a legend to make referring to the machine learning method easier; Indicate title on the graph
- In Section 3.5, it was mentioned that the importance of each variable was calculated, yet no explanation/description as to how it was done.
- The discussion in Section 4 mentioned previous studies utilising imbalanced methods, these could be described earlier in the Introduction or Literature Review in the paper
- Check the appropriate usage of the words "besides" and “thus”.
- Line 161, why RF used as reference.
- Line 170 why logistic regression was used as a reference.
- Line 163 how did you calculate p-value; which method was used? It seems you calculated p-value comparing two numbers how? doesn't make sense?
Author Response
Dear reviewer,
Thanks for your careful review and insightful comments toward our manuscript. Based on your suggestions, we carefully revised our manuscript. And the questions were answerd point by point. Please see the attachment. Once again, thanks very much!

Reviewer 2 Report
The manuscript titled “Machine Learning-Based Prediction of Stroke with Imbalanced Data: The Chinese Longitudinal Healthy Longevity Study” used data balancing techniques and machine learning approach to predict stroke in a Chinese population. The results was moderate in predicting stroke using patients’ demographic, life styles, and clinical variables with area under curve at around 70%.
This is an interesting study, however, there are error and lack of clarity on study methods that needs to be addressed as following:
- Introduction session gave very brief description on the imbalanced data in prediction studies, it needs to discuss the strength and weakness of each of the three data balancing techniques such as random over sampling (ROS), random under sampling (RUS), and synthetic minority over sampling (SMOTE) techniques and how these methods were used in other studies to improve predictions.
- Similar to the data balancing techniques, the authors should discuss the pros and cons for the three types of machine learning techniques used in prediction studies and why these methods were chosen to predict stroke.
- In study methods session, the authors should state the selection criteria for the study population. Is this a cross sectional study or a longitudinal study?
- The study outcome variable is patient self-reported stroke. How was this variable collected? Is it through survey or questionnaire? Age variable was the age that patient had stroke or the age was at the time of survey?
- The authors reported that there were total 2439 patients in 2012 and 1109 (45%) were excluded due to missing data. Sensitivity data analysis should be used to compare patients’ characteristics in the study sample and those who were excluded to assess if missing was random. Also, imputation methods should be consider for data missing.
- In results session, there is an error in line 126, it should be men experiencing more stroke than women.
- For table 1, I’d suggest to report all stratum of the categorical variables. For example, for sex, both frequency and row percentage for women and men should be reported so readers can clearly see the prevalence of stroke were 6.76% for men and 3.25% for women. Please use the raw p-value of two decimal point instead of using <0.05 or >0.05.
- As of the results of sensitivity, specificity, accuracy, and AUC obtained from different machine learning techniques, is there a statistical test can be used to test if there is significant difference among these three tests?
- This study is to predict stroke, besides report AUC, are there other measurement such as risk ratio or odds ratio can be reported? Authors can do a multivariable logistic regression modeling for this study, and compare the AUC with those obtained using machine learning methods to see if there is any advantage of using machine learning methods.
- In discussion session, please state what kind of study population can this results be generated to.
Author Response

(The authors gave the same response as above.)

Round 2
Reviewer 1 Report
The authors have improved their article, however, still authors have not fully implemented my previous suggestions. In the Authors response to the reviewer, authors should provide line numbers and/or page numbers to easily find the changes.
- Authors introduce new paragraph in the Introduction line 69 to 84 but did not cite and discussed the relevant work dealing with imbalance data that I pointed before doi: 10.3390/data4030129
- Section 2.5 can be improved by providing a mathematical formula of how p-values were calculated.
- No description of how p-values in Table 1 were calculated.
- Table 3 rank of each predictor should be compared with the p-values of association of each predictor against the outcome.
Author Response
Thanks for the reviewer's comments and suggestions. Based on your advice, we modified our manuscript. And the questions were answered point by point. Please see the attachment.
Once again, thanks very much.

Reviewer 2 Report
The authors made changes according to reviewer's comments. The paper is improved and can be accepted for publication.
Author Response
Thanks for the reviewer's careful review and comments. We are grateful for the time you have spent for reviewing our manuscript.